# Colorimetric Assaying of Exosomal Metabolic Biomarkers

**DOI:** 10.3390/molecules28041909

**Published:** 2023-02-16

**Authors:** Evelias Yan, Garima Goyal, Umit Hakan Yildiz, Bernhard O. Boehm, Alagappan Palaniappan

**Affiliations:** 1Center for Biomimetic Sensor Science, Nanyang Technological University, Singapore 637553, Singapore; 2School of Materials Science and Engineering, Nanyang Technological University, Singapore 639798, Singapore; 3Interdisciplinary Graduate School, Nanyang Technological University, Singapore 637335, Singapore; 4Department of Chemistry, Izmir Institute of Technology, 35430 Izmir, Turkey; 5Lee Kong Chian School of Medicine, Nanyang Technological University, Singapore 308232, Singapore

**Keywords:** exosome, exosomal biomarkers, glycated proteins, polythiophene, mir21, HBV-DNA

## Abstract

Exosomes released into the extracellular matrix have been reported to contain metabolic biomarkers of various diseases. These intraluminal vesicles are typically found in blood, urine, saliva, breast milk, cerebrospinal fluid, semen, amniotic fluid, and ascites. Analysis of exosomal content with specific profiles of DNA, microRNA, proteins, and lipids can mirror their cellular origin and physiological state. Therefore, exosomal cargos may reflect the physiological processes at cellular level and can potentially be used as biomarkers. Herein, we report an optical detection method for assaying exosomal biomarkers that supersedes the state-of-the-art time consuming and laborious assays such as ELISA and NTA. The proposed assay monitors the changes in optical properties of poly(3-(4-methyl-3′-thienyloxy) propyltriethylammonium bromide) upon interacting with aptamers/peptide nucleic acids in the presence or absence of target biomarkers. As a proof of concept, this study demonstrates facile assaying of microRNA, DNA, and advanced glycation end products in exosomes isolated from human plasma with detection levels of ~1.2, 0.04, and 0.35 fM/exosome, respectively. Thus, the obtained results illustrate that the proposed methodology is applicable for rapid and facile detection of generic exosomal biomarkers for facilitating diseases diagnosis.

## 1. Introduction

Exosomes are functional nanosized membrane vesicles [1] that carry biologically active molecules, such as proteins [2], lipids [3], nucleic acids [4,5], microRNA, etc. They are involved in processes such as immune response [6] and intercellular communication [7]. The use of exosomal contents as biomarkers in its infancy state has acquired significant interest owing to its specificity with the cellular processes. Exosomes have been discovered in blood [8], urine [9], saliva [10], breast milk [11] and amniotic fluid [12]. The content of exosomes varies significantly based on the originating cell type or its state and therefore they can be utilized as biomarkers for prognosis of metabolic diseases, such as inflammatory complications [13], cardiovascular diseases [14], neurodegenerative diseases [15,16] and tumors [17,18]. The other advantages, such as the abundance of exosomal biomarkers and the recent developments in extraction of exosomes from biological fluids indicate the huge potential for exosomal biomarker-based diagnosis [19].

Recently, various techniques to characterize exosomes qualitatively and quantitatively have been reported. These methods could be broadly categorized into two groups: external characterization to determine size and morphology, and internal characterization to identify membrane proteins and lipid contents [20,21]. The methods for external characterization include: (1) nanoparticle tracking analysis to measure concentration and size [22,23]; (2) dynamic light scattering to measure particle size via a monochromatic laser beam passing through a suspension of exosomes [24]; (3) atomic force microscopy to scan the surface of exosomes [25]; (4) transmission electron microscopy (TEM), scanning electron microscopy (SEM), and cryo-electron microscopy (CryoEM) to visualize the size and morphology of exosomes [22,24]; (5) tunable resistive pulse sensing (TRPS) to determine concentration of exosomes [26]; (6) digital holographic tomography (DHT) to visualize three-dimensional images without labelling [27]; and (7) surface enhanced Raman spectroscopy for qualitative analysis of extracellular vesicles [28].

As for internal characterization, various analytical assays have been developed for quantitative detection of exosomal biomarkers. The common methods are western blotting [29], ELISA [30], mass spectrometry [31], qRT-PCR [24,32] and microarray [33] for the detection of exosomal proteins and nucleic acids. Although these methods are sensitive and specific, they require expensive and sophisticated instrumentation, trained personnel, and are time consuming. Therefore, there is a need to develop next generation assays that can quantify exosomal biomarkers with high sensitivity, selectivity, and ease that overcome the limitations of the above-mentioned methods, while holding potential to be translated from laboratory-based detection into a portable format for on-site analysis or field use.

Due to simplicity and ease of use, colorimetric assays possess the potential to be translated into rapid and cost-efficient portable sensors. Conventional optical reporters, such as dyes and pigments, possess sensitivity limitations owing to smaller extinction coefficients, chemical instability at high temperature, toxicity, and lower resolution, limiting their optical performance. Therefore, the development of nanomaterials, such as magnetic nanoparticles [34], gold nanoparticles [35], graphene oxide [36], carbon nanotubes [37] and conjugated polymers [38] has been explored for optical assaying to address the limitations of conventional dyes. CPs in particular are reported to be very sensitive for protein/nucleic acid assaying, owing to their complexation capabilities with recognition molecules, such as aptamers, that specially interact with target analytes.

Herein, we report an optical method for detection of exosomal biomarkers. Optical properties of poly(3-(4-methyl-3′-thienyloxy) propyltriethylammonium bromide) (PT) are monitored upon interacting recognition elements such as aptamer [39] and peptide nucleic acid (PNA) [40] in the presence or absence of target biomarkers. Herein, an approach for facile assaying of microRNA, DNA, and advanced glycation end products (AGE) in exosomes isolated from human plasma is demonstrated. The methodology yielded detection limits of ~1.2, 0.04, and 0.34 fM/exosome, respectively, illustrating the potential for analysis of state of the metabolic diseases and for facilitating diseases diagnosis.

## 2. Results and Discussion

For assaying, the exosomes are extracted from human plasma and lysed using 0.1% Triton-X 100 to yield free target molecules, which are then allowed to interact with PNA or aptamers. PT is subsequently added to observe color changes, as shown in Figure 1. In this study, mir21 (a microRNA sequence associated with various cancers), HBV-DNA (a DNA biomarker of hepatitis), and glyceraldehyde-derived AGE (a metabolic disease AGE biomarker) are utilized as model biomarkers to validate the proposed methodology. For the nucleic acid assaying (mir21 and HBV-DNA), PT without PNA forms a duplex with nucleic acids (PT-mir21/PT-HBV-DNA) through electrostatic interactions between PT and nucleic acids, leading to a color change from orange (PT color) to purple (color of PT-nucleic acid) and to a fluorescence quenching depending on the concentration of the nucleic acids. In contrast, PT with complementary PNA to mir21/HBV-DNA forms a triplex upon interaction with target nucleic acid, inhibiting the formation of the PT-mir21/HBV-DNA duplex, and therefore does not yield significant color and fluorescence changes. The orange color (PT-PNA-mir21/HBV-DNA) and the significantly different purple color (PT-mir21/HBV-DNA, without PNA) indicates the presence or absence of mir21/HBV-DNA in the sample, respectively, enabling colorimetric detection of nucleic acids. The PNA sequence that is complementary to the target nucleic acid ascertains the specificity of the assay.

Similarly, for AGE assaying, in the absence of AGE, a duplex is formed between PT and Apt due to electrostatic interactions between the positively charged PT and negatively charged Apt, causing a color transition in the PT from orange to purple. On the contrary, in the presence of AGE, the electrostatic forces between PT and Apt are reduced due to the competitive interaction of Apt with AGE, and do not yield significant color and fluorescence changes. Thus, the orange color (PT-Apt-AGE) and significantly different purple color (PT-Apt, without AGE) correspond to the presence and absence of AGE, respectively, thereby enabling the direct visual detection of AGE. Again, the aptamer sequence that is complementary to the AGE ascertains the specificity of the assay.

### 2.1. Characterization of Exosomes before and after Lysis with 0.1% Triton X-100

The exosome samples before and after lysis were characterized based on their morphology and size, using cryo-electron microscopy (CryoEM). CryoEM allows the visualization of a broad spectrum of extra-cellular vesicles of varying size, morphologies with lipid bilayers, and the internal structures of vesicles. Furthermore, the preparation of samples for CryoEM analysis does not require any staining or chemical fixation and samples are directly drop-casted on an EM grid to be vitrified and visualized. Thus, it preserves the exosomes and enables observation of biological structures in a vitrified near-native state [41,42,43]. As shown in Figure 2a, the digital image from the exosome sample before lysis shows a spherical vesicle of approximately 90 nm, as measured from the scale bar of the digital image. This indicates that the vesicles in the exosome sample before lysis remain intact. Conversely, in Figure 2b and Appendix A, the digital images from the exosome after lysis show a random size distribution of particles. This shows that the vesicles rupture upon lysing with 0.1% Triton X-100.

Dynamic light scattering (DLS) analysis shows that the hydrodynamic size of the exosomes before lysis is approximately 120 nm with a poly-dispersity index (PDI) of 0.2, as shown in Figure 3a. However, in Figure 3b, a wider size distribution of the vesicles with a PDI of 0.8 was observed upon lysis, indicating that the vesicles have been ruptured. In Figure 3c, the DLS analysis shows that the Triton X-100 that was used as a lysing reagent at a concentration of 0.1%, forms vesicles of approximately 10 nm, which is smaller than that of the exosomes with a narrow size distribution.

### 2.2. The Influence of Tween 20 (T20)

The exosomal cargos could potentially interfere with the fluorescence properties of the cationic PT; for example, the negatively charged phosphate backbone of exosomal nucleic acids might interact with PT via electrostatic attractions, thus affecting the fluorometric and colorimetric response of the assays. Therefore, the interference from the exosomal cargos should be minimized for the development of a sensitive and robust colorimetric assay for the detection of target exosomal biomarkers. In this study, we have systematically analyzed the effect of different buffer solutions on the assay response. As illustrated in Figure 4a, the digital images of vials A–C in MilliQ water show that the color changes from light yellow to bright yellow, with a decreasing concentration of exosomes in the vials. This is further evident in the fluorescence spectra in Figure 4d, whereby the fluorescence intensity increases from vial A to vial C. Similar responses are observed in Figure 4b,e when the exosomes in varying concentrations are added to PBS. As such, there is a possibility that the exosomes are interfering with the colorimetric response of the PT. Interestingly, in the presence of T20, the digital images of vials G–I did not yield any change in color, which is further evident by the overlapping fluorescence spectra of all three vials, as shown in Figure 4c and Figure 4f, respectively. Thus, T20 was added to the PT solution at a concentration of 0.01% for all the colorimetric assays.

### 2.3. Presence of mir21 Exosomes Extracted from Cancerous Cells

The RNA melting curve analysis in Figure 5 shows that exosomes contain mir21 in their content, as described by previous reports [44]. Upon 25 cycles of amplification of extract from exosomal content, genetic material yields a melting point at approximately 65–67 °C, which corresponds to mir21. Although, there are slight variations in the melting temperature, which may be due to the higher molecular weight of nucleic acid residues, the major product is determined to be mir21, considering that the amplification was carried out in the presence of the complementary PNA sequence. This observation ascertains that mir21 is in abundance in exosome samples utilized in this study.

### 2.4. mir21 Assay

Figure 6a illustrates the colorimetric responses of exosome samples to various concentrations of mir21 (vials: B, 4 μM; C, 2 μM; D, 1 μM; E, 500 nM; F, 250 nM). With PNA1 (complementary PNA sequence to mir21, vial A, control), PT-T20-PNA1-mir21 yields a yellowish-orange color. The color changes from dark orange with 4 μM of mir21 (vial B). As the concentration of mir21 decreases (from 4 μM (vial B) to 250 nM (vial F)), the color changes from dark orange to light orange as shown in Figure 6a. ΔE values, a parameter to quantify the color changes, were subsequently calculated to ascertain the colorimetric response of the assay. A ΔE-value of 2 is considered as the threshold for differentiating two colors [45]. Figure 6c shows that the ΔE values range from 5 to 15 from vial B to F, with respect to vial A, indicating that the color difference between PT-T20-mir21 and PT-T20-PNA1-mir21 can be visually distinguished, whereas vials E and F yield similar ΔE values. The ΔE values for each concentration of mir21, averaged for three individual experiments (*n* = 3) and plotted as a linear function of 4 μM to 250 nM exosomal mir21 concentrations, yield a colorimetric limit of detection (LOD) of 3.4 fM/exosome using the 3σ/S approach with an assay time of 30 min.

To further validate the assay, the fluorescence spectra were recorded for the corresponding colorimetric responses. In the absence of PNA1, the steady-state fluorescence spectroscopy displays a substantial fluorescence quenching in emission and a red shift to 620 nm in PT-T20-mir21, with respect to the baseline, PT-T20-PNA1-mir21 that has a peak maximum at 600 nm as shown in Figure 6b. This indicates a duplex formation between PT-T20 and mir21. Moreover, it could be observed that the fluorescence intensity increases gradually with the decreasing concentration of mir21 and approaches saturation at 250 nM. The fluorescence maxima of each concentration of mir21 were then plotted against the various concentrations of mir21, which shows that the fluorescence intensity increases gradually from 500 nM to 4 μM of mir21 (Figure 6d). The maxima values for each concentration of mir21, averaged for three individual experiments (*n* = 3) and plotted as a linear function of 4 μM to 250 nM exosomal mir21 concentration, yielded an LOD of 1.2 fM/exosome using the 3σ/S approach, which concurs with the LOD obtained using colorimetric analysis.

The assay response for exosomal HBV-DNA is then performed using the protocol adopted for mir21 assay with a PNA sequence, PNA2, that is complementary to HBV-DNA (Appendix A). Upon colorimetric, ΔE, and fluorometric response analyses, a colorimetric LOD of 0.12 fM/exosome and a fluorometric LOD of 0.04 fM/exosome were evaluated. In addition to nucleic acid assaying, the proposed assay was utilized to detect exosomal AGE with the use of an aptamer (Apt) as the recognition element (Appendix A). The colorimetric responses, ΔE analysis, fluorescence spectra, and calibration curve were then established. Appendix A shows the colorimetric responses of various concentrations of AGE (vials: B, 974 nM; C, 877 nM; D, 779 nM; E, 682 nM; F, 584 nM). With Apt 1 (vial A, control), PT-T20-Apt yields a purple color, indicating fluorescence quenching. As the concentration of AGE decreases (from 974 nM (vial B) to 584 nM (vial F)), the color changes from orange to purple (Appendix A). The ΔE value (Appendix A) illustrates that the colorimetric responses for PT-T20-Apt and PT-T20-Apt-AGE can be visually distinguished. The colorimetric limit of detection was calculated based on the 3σ/S approach, which yields an LOD of ~0.63 fM/exosome. Furthermore, the fluorescence spectra show that as the concentration of AGE decreases, the fluorescence intensity decreases, as illustrated in Appendix A. The maxima of each AGE concentration were then plotted as a linear function, which illustrates that the fluorescence intensity decreases gradually from 974 nM to 584 nM of AGE (Appendix A), yielding an LOD of ~0.34 fM/exosome using the 3σ/S approach. Thus, the proposed approach ascertains that colorimetric detection of the different exosomal biomarkers is feasible.

### 2.5. Comparative Analysis of LODs and Scope for Improvement

In this present work, assaying of exosomal markers, such as RNA, DNA, and protein molecules, with LOD in the range of fM/exosome has been demonstrated. The LODs obtained via the proposed approach is compared with detection of exosomal markers in other biological matrices and shown in Table 1.

The proposed assay is generic in nature and possesses the potential to be customized for the detection of disease-specific biomarkers upon utilization of appropriate recognition elements. Although exosomes are being widely considered as diagnostic vehicles, temperature-dependent stability of exosomes is one of the major concerns that limits practical applications. The complex constitution of exosomal extract could also influence the accurate and precise detection of biomarkers. Hence, the accuracy of the proposed assay relies on the efficiency of the exosome extraction protocol and the obtained yield. Substantial research efforts are therefore required for the deployment of exosomal assays for clinical applications.

## 3. Materials and Methods

### 3.1. Materials and Chemicals

The required chemicals for PT synthesis were obtained from Sigma-Aldrich (Singapore) and used without further purification. The polyoxyethylene (20) sorbitan monolaurate solution (10% T20) and 4-(1,1,3,3-tetramethylbutyl) phenylpolyethylene glycol (triton X-100 (TX100)) were obtained from Bio-Rad Laboratories (Singapore). The phosphate-buffered saline (10× PBS, pH 7.2) was purchased from Thermo Fisher Scientific. MicroRNA 21 and HBV-DNA sequences were obtained from IDT (Singapore), and AGE was obtained from MBL Life Science (Circulex, Nagano, Japan). The aptamer for AGE (Apt 1, 5′-TGT AGC CCG AGT ATC ATT CTC CAT CGC CCC CAG ATA CAA G-3′) was synthesized by IDT (Singapore) with PAGE purification. Peptide nucleic acid 1 (PNA 1: N-TCA ACA TCA GTC TGA TAA GCT A–C) and peptide nucleic acid 2 (PNA 2: N-AAT ACC ACA TCA TCC ATA TAC) were purchased from Panagene (Daejon, Korea).

### 3.2. Extraction and Purification of Exosomes from Human Plasma and Lysis of the Exosomes

The extraction and purification of exosomes were carried out based on reported protocols with necessary modifications [46,47,48]. All blood samples were taken in the fasting state. Subjects with elevated blood lipids (elevated total cholesterol and/or triglycerides) were excluded. Venous blood samples used in this study were drawn into BD Vacutainer^®^ Lithium Heparin tubes. In brief, after collection of the blood sample, the tube was centrifuged immediately at 1800× *g* for 6 min at room temperature (centrifugation step 1). Heparin plasma was retrieved from the tubes without touching the buffy coat. This centrifugation stage was followed by a 2-step centrifugation at 2000× *g* for 30 min at 15 °C (centrifugation step 2). Each time, two-thirds of the plasma was transferred to a new centrifugation tube. After the 2nd round of centrifugation, the top two-thirds of the heparin plasma was aliquoted into cryotubes and instantly stored at −80 °C before further use. After thawing, samples were ultra-centrifuged at 135,000× *g* for 2 h (density gradient ultracentrifugation). Only the elution fraction with a low signal of lipoprotein markers (Apo A and Apo B) based on a proteomics analysis and an enrichment of EV markers (CD9/CD63/CD81) was used for further studies. Subsequently, these aliquots containing an enrichment of exosomes were then lysed with 0.1% Triton X-100, which was used for all the colorimetric assays in this study.

### 3.3. Presence of mir21 Exosomes Extracted from Cancerous Cells

Exosomes were prepared using the ultracentrifugation method: a culture medium of colon cancer cell line HT29 was centrifuged at 15,000 rpm for 55 min to eliminate the cell debris. The supernatant was filtered through a 0.20 μm filter and centrifuged at 180,000× *g* for 45 min + 45 min with a 15 min interval between consecutive runs. The pellets were treated with phosphate-buffered saline (PBS) and then resuspended in PBS as exosome-enriched fractions. These fractions treated with 1 mL Trizol LS (Invitrogen), the aqueous phase was separated and mixed with 500 μL ethanol, and RNAs were collected using RNeasy column (Qiagen).

### 3.4. Synthesis of PT

The polythiophene (PT) used in this study was synthesized as reported previously [30]. T20 was added to the PT solution at a concentration of 0.01% for all the colorimetric assays.

### 3.5. Assay Preparation

For the miRNA assay, 10 μL of various concentrations of miRNA was spiked and diluted 10 times and exosome samples were added to 10 μL of 5 μM complementary PNA 1 and to 10 μL of PBS (without PNA), incubated for 30 min, followed by the addition of 30 μL of 60 μM PT-T20 solution. For the HBV-DNA assay, 10 μL of various concentrations of HBV-DNA-spiked exosome samples were added to 10 μL of 5 μM complementary PNA 2 and to 10 μL of PBS (without PNA), incubated for 30 min and followed by the addition of 30 μL of 60 μM PT-T20 solution. For the protein (AGE) assay, 15 μL of various concentrations of AGE-spiked, 10 times-diluted exosome samples were added to 5 μL of 1 μM aptamer (Apt 1) and incubated for 30 min. Subsequently, 30 μL of 120 μM PT-T20 solution was added.

### 3.6. RGB Analysis of the Colorimetric Assays

The RGB analysis of the colorimetric assays was conducted similar to our previous reports with some modifications [49]. Briefly, the vials were placed under a UV illumination zone (portable UV lamp, at 365 nm, 8 W) for the visualization of the change in color of the assay. Digital images were then captured using an iPhone 11 mobile phone camera positioned at a fixed distance and in fixed lighting conditions. The digital images were then imported into a computer as raw jpeg files for image processing. Five regions of interest with a maximum coverage of the image, each of size 5 × 5 pixels, were captured for the vials for RGB analysis using ImageJ (image processing software) to determine colorimetric responses of the assay. The RGB values for each concentration of analyte were averaged for 3 individual experiments (*n* = 3). Subsequently, ΔE was calculated from the obtained RGB values to determine the colorimetric response of the assay. In this work, ΔE values were obtained using the CIE2000 algorithm [50] with the ColorMine online calculator [45,51].

### 3.7. Fluorescence Spectra Analysis of the Assays

Steady-state fluorescence spectra of the samples were measured using an Infinite M200 Pro Tecan plate reader. Fluorescence spectra were obtained with an excitation wavelength of 480 nm.

### 3.8. Cryogenic Electron Microscopy

The preparation of exosome samples for imaging under a cryogenic electron microscopy was performed using the method reported previously [52]. Electron microscope grids were first coated with a holey carbon film and then glow discharged. A total of 4 μL of the exosome samples (before and after lysis with 0.1% triton X-100) were drop-casted onto a grid at 99% humidity, blotted with a filter paper, and plunged into liquid ethane (Vitrobot, FEI Company, Singapore). Cryo-grids were imaged using a FEG 200 keV transmission electron microscope (Arctica, FEI Company) equipped with a direct electron detector (Falcon II, FEI Company). 

### 3.9. Dynamic Light Scattering

Samples of exosomes before and after lysis with 0.1% Triton X-100, and 0.1% Triton X-100, were characterized using a Malvern Zetasizer Nano ZS, Malvern Instruments Ltd. (Worcestershire, UK). A total of 500 µL of each sample was pipetted into a disposable cuvette, subjected to particle size analysis at a 173 °C and a backscatter angle at 25 °C for DLS measurements. Every measurement was an average of 10 runs recorded 3 times. The hydrodynamic size and distribution of the samples were obtained and discussed in Section 2.1.

## 4. Conclusions

This study demonstrates facile assaying of microRNA, DNA, and advanced glycation end products in exosomes isolated from human plasma. The proposed methodology offers a unique strategy for the detection of a broad range of biomarkers (microRNA, DNA, and proteins) in exosomes with competitive LODs. Exosomal biomarker assays are potentially a more relevant approach for diagnosis than circulating nucleic acids/proteins in plasma, as these markers are precise indicators of the state of originating cells. LODs of 1.2 fM/exosome, 0.04 fM/exosome, and 0.34 fM/exosome, and colorimetric detection limits of 3.4 fM/exosome, 0.12 fM/exosome, and 0.63 fM/exosome, for mir 21, HBV-DNA, and AGE, respectively, were achieved. Thus, the proposed methodology enables rapid detection of generic exosomal biomarkers for analysis of the state of diseases, thereby facilitating disease diagnosis.

## Figures and Tables

**Figure 1 molecules-28-01909-f001:**
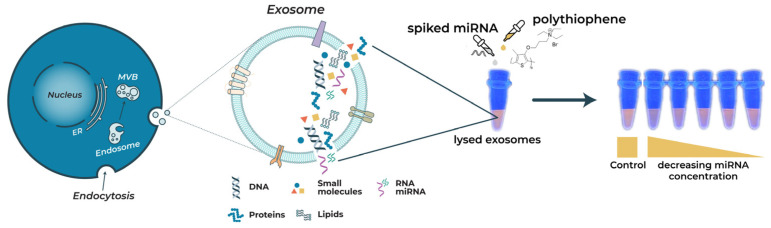
Exosome extraction from human plasma and lysis to release target molecules. Colorimetric responses of PT upon the addition of PNA and various concentrations of the mir21-spiked exosomes.

**Figure 2 molecules-28-01909-f002:**
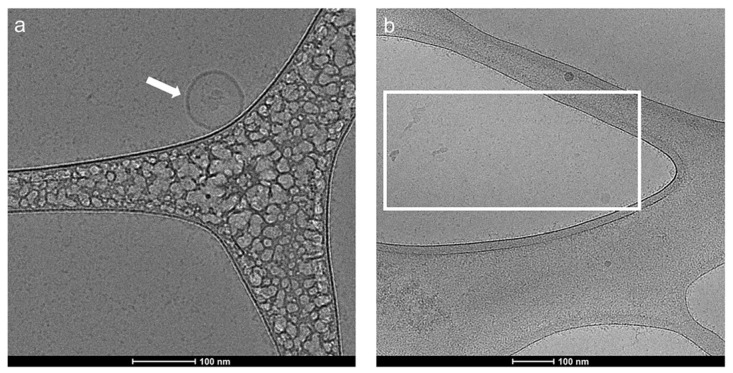
Electron microscopy images of exosome sample (**a**) before lysis (as indicated by the white arrow that shows an exosome) and (**b**) after lysis (as indicated in the white box whereby the vesicles have been ruptured).

**Figure 3 molecules-28-01909-f003:**
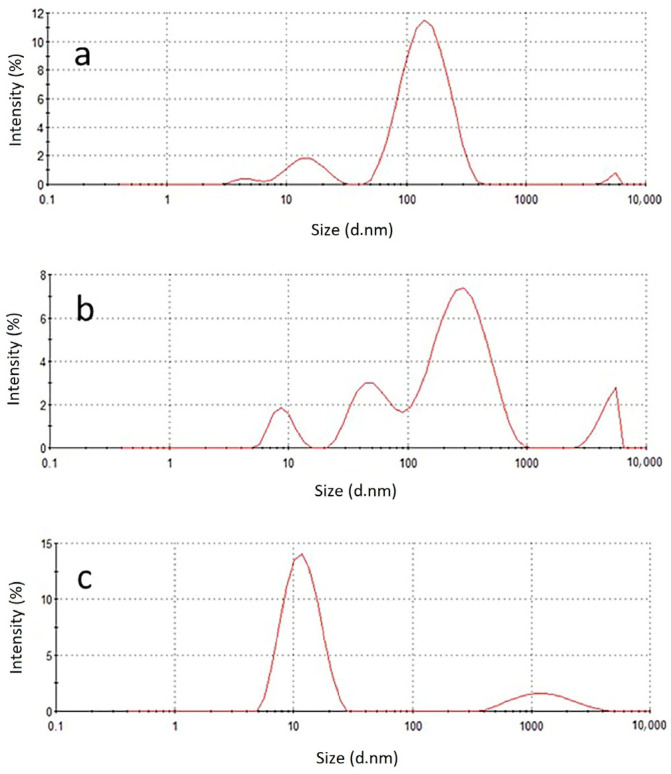
Size distribution spectrum of (**a**) exosome sample before lysis, (**b**) exosome sample after lysis, and (**c**) 0.1% Triton X-100.

**Figure 4 molecules-28-01909-f004:**
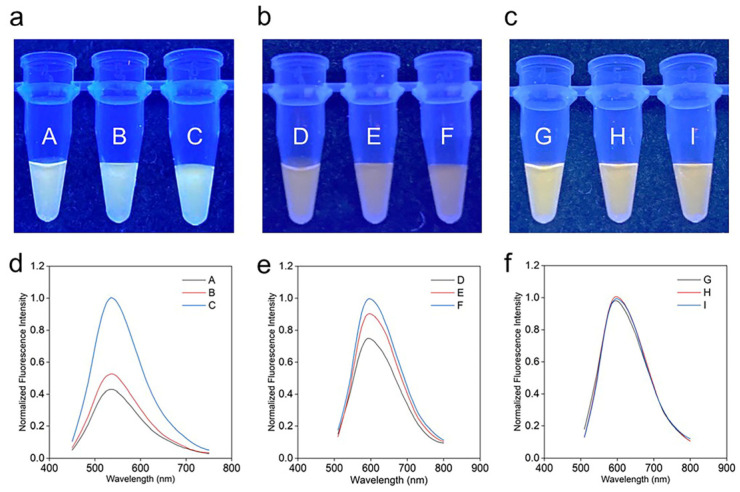
Colorimetric responses (digital images) of varying amounts (1×, 10×, 0×) of (**a**) exosomes with PT in MilliQ in Vials A–C, (**b**) exosomes with PT in PBS in Vials D–F, and (**c**) exosomes with PT-T20 in PBS in Vials G–I, and their corresponding fluorescence spectra (**d**–**f**), respectively.

**Figure 5 molecules-28-01909-f005:**
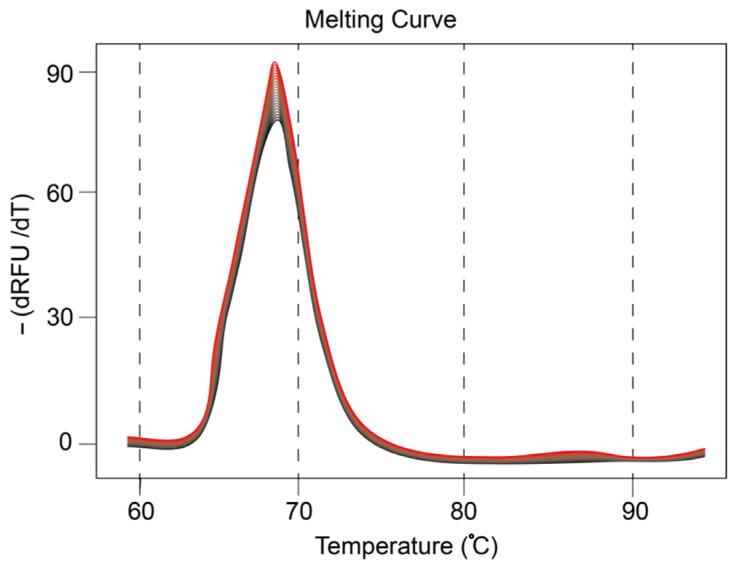
Melting curve of mir21 expression (20 cycles).

**Figure 6 molecules-28-01909-f006:**
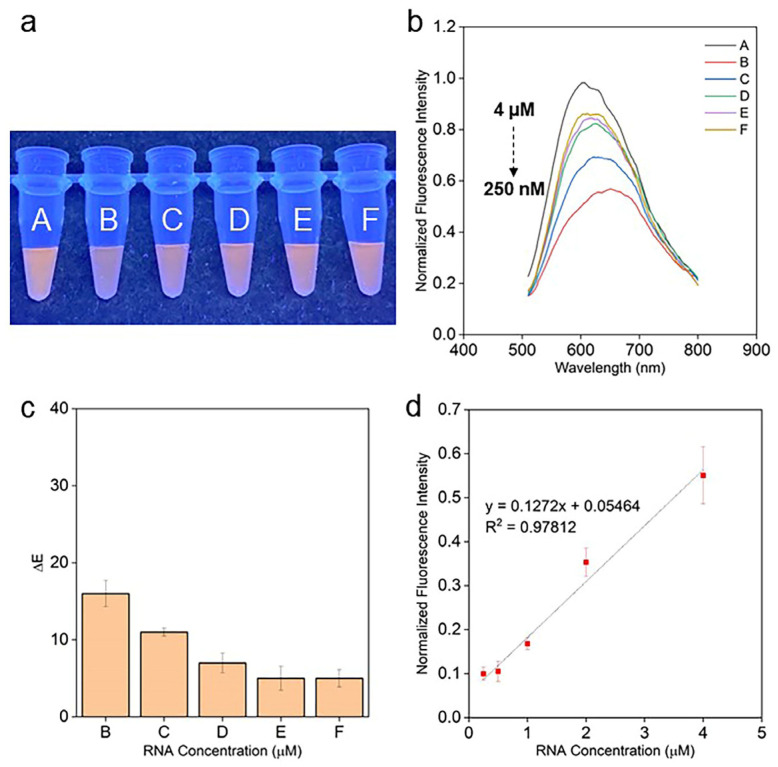
mir21 assay (**a**–**d**): (**a**) digital images of vials containing (A) PT-T20-PNA1-1 μM mir21 (control), and (B–F) PT-T20 with 4 μM, 2 μM, 1 μM, 500 nM and 250 nM mir21 and, (**b**) corresponding fluorescence spectra, (**c**) ΔE values calculated with respect to vial A and (**d**) linear plot of mir21 concentration versus normalized fluorescence intensity.

**Table 1 molecules-28-01909-t001:** Comparison with optical assays for biological biomarkers in biological fluids.

Biomarker	Reference LOD	Reference Sample	Reported LOD
miRNA	2 nM [31]	Plasma	1.2 fM/exosome
HBV-DNA	1 nM [30], 2 nM [31]	Urine, plasma	0.04 fM/exosome
AGE	850 pM [30]	Urine, plasma	0.34 fM/exosome

## Data Availability

Not applicable.

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
