# Peer review of "Colorimetric Assaying of Exosomal Metabolic Biomarkers"

_molecules, 2023, doi:10.3390/molecules28041909_

Round 1
Reviewer 1 Report
The article is interesting. However, there are certain concerns that has to be addressed to make the article better.
Those comments are attached
Comments
Minor comments
Line 32 & 33: Check grammer
Line 47: proteins to qRT-PCR and, microarray
Major comments
2. Results and discussion
Question 1: How can it be confirmed that it is the mir27/HBV DNA that is specifically binding to the PT, when non complexed with PNA. It could also be other micro RNAs or nucleic acid fragments that is being bound. I would advice the authors to perform a qRT-PCR to determine the specifity of mir21/HBV DNA binding to PT. This could also state the concentration of mir21 been bound.
Question2: Similarly, to what is stated above, how is it possible to confirm the it is AGE protein that is competing for binding to PT. It could be any other protein as well. This specific binding /competition must be verified by other methods as well like western blot or Mass spec or so.
2.1. Line 112. Here the authors are making use of CryoEM to determine the size and morphology of the exosomes. Why was cryoEM used specifically. This has to be discussed briefly with references.
Line 114: The authors state the size of exosomes was 90 nm. How did the authors determine the size?
Line 120: In figure 3b, there is awider distribution of vesicles, but the predominant size observed was around 500 nm. How does this size difference indicate that the vesicles were ruptured as claimed by the authors.
2.3. The melting curve analysis states that mir21 is present in the exosomes. But, how can it rule out the possibility of other microRNAs /nucleic acids been absent? Also, how do the authors claim that it was mir21 present in abundance as sated in line 161
Line 183: Is it a duplex formation or binding. Normally duplex is formed with two complimentary sequences. How a duplex can form between PT and mir21?
3.2 and 3.3: Here the authors describe the extraction of exosomes from plasma and cancerous cells. Is the protocol described a standard method for exosome extraction? If so, please cite the reference

Reviewer 2 Report
The manuscript of Evelias Yan Hui Xin et al. is concerning the examination of the novel colorimetric assay for the detection of exosomal metabolic biomarkers. It is an interesting approach, since there are perspectives of the use of exosomes or generally extracellular vesicles/EVs (particularly microRNAs and cargo content) for as disease biomarkers. Moreover, EVs have attracted much interest in the past decade due to their potential utility as circulating biomarkers for cancer.
The manuscript is within the scope of the Molecules Journal. The logical structure and organization of the manuscript are fine. The choice of references is fine, but some important recent literature on modern exosome phenotyping techniques is missing, and this should be corrected. In my opinion, before the acceptance of the submitted manuscript for publication, additional explanations and significant corrections have to be made.
Major Comments:
1) In the Introduction section, the authors did not describe any recent advances in the measurement techniques focused on the EV phenotyping including SERS, digital holographic tomography or, nanoparticle tracking analysis, AFM, quantitative polymerase chain reaction, dynamic light scattering, etc. (see some examples below)
https://doi.org/10.1002/adfm.202010296
https://doi.org/10.1016/j.ajpath.2021.08.005
https://doi.org/10.1002/mnfr.201500222
https://doi.org/10.1039/D0NR07349K
https://doi.org/10.1016/j.jconrel.2018.08.035
https://doi.org/10.1038/s41467-020-14344-7
2) I wonder why the authors mention miRNA-21 among potential biomarkers, since it was already reported (https://doi.org/10.1186/s40364-021-00272-1 ) that ‘miR-21 cannot be considered a specific biomarker for any disease if it is a biomarker of many diseases. While its levels may genuinely vary across bodily fluids in disease states, these variations have no specificity. ‘ Therefore, I wonder why the authors decided to consider such nonspecific miRNAs as biomarkers of diseases.
3) The effectiveness of the proposed method is entirely on the efficiency of the isolation of exosomes from body fluids. There are a wide range of techniques that can be used for isolating EVs from biofluids, which have been reviewed extensively. A key area of difference between recent studies is the biofluid used to isolate EVs. Although EVs have been isolated from almost all bodily fluids, the main sources commonly used include urine and blood (both serum and plasma). One of the most abundant sources of EVs is blood ( as in the submitted manuscript); however, the most challenging aspects of the isolation of EVs from blood are that it contains lipoproteins and chylomicrons, which overlap in size and density with EVs and cannot be completely removed by conventional isolation methods. In addition, the amount, purity and heterogeneity of EVs from blood are influenced by sample collection, handling, storage conditions, stability, anticoagulants, volume of blood collection, time of blood collection, and the age, sex, disease state and fed/fast status of the animal/patient. Are the authors sure that they examined only isolated EVs?
4) The growing interest in EV makes the technical standardization extremely important, because many methodologies can be used to isolate and analyze EV. Different isolation procedures, a variety of techniques used to purify RNA, can significantly affect extracellular RNA sequencing and profiling, making the results unclear. However, the procedure of the EVs isolation and purification is briefly described in this manuscript. Provide a more extended description of the procedures and methodology.
5) The method of analysing the images demonstrating the colour change of the test was described in a very cursory manner. The authors are asked to describe the individual steps of the image analysis in detail.
6) I do not fully understand which light source was used to induce fluorescence. Initially the authors wrote about illuminating the samples with UV light (365 nm) for the visualisation in the change in colour of the assay, and then that fluorescence spectra were recorded for 480 nm? What is the reason for this difference? Why did not decide to use one light source? The fluorescence spectra can be different for different excitation wavelengths.
7) Why was UV-A used if it can induce stress proteins? Furthermore, it can induce the generation of free radicals that damage DNA. Could this affect the results obtained?
8 )What does the statement "RGB values" mean, i.e. the average pixel intensity of the ROI area for all channels?
9 )Why did the authors choose not to divide the images into individual R, G, and B channels? After all, from the results presented in Fig.4, we can see that for exosomes with PT in MilliQ in Vials A - C, the maximum changes in fluorescence intensity are obtained in the green channel (500-560 nm), while for exosomes with PT in PBS in Vials D - F and PT-T20 in PBS in Vials G - I in the red channel. The blue channel does not contribute any information related to fluorescence at all.
10 )If the authors recorded the image of the vials, why did they select only five regions of interest (ROI) of 5 x 5 pixels for analysis, and not segment the area of the vial and determine the average intensity for the entire area occupied by the test sample? What percentage of the sample area was surveyed with this choice of ROIs? Such an approach raises my doubts about the representativeness of the results obtained.
11) What was the criterion for selecting ROIs?
12) Fig.6d: Provide verification of the quality of the fit
13) In the Materials and Methods section, there is no description of the Dynamic light scattering examination of the samples, the measurements system, measurements protocols even if the results of this examination were presented in the Results section. Provide missing information.
14) In my opinion, the section on the research methodology and the measurements used is described too briefly. A lot of important information is missing.
15) How was the delta-E parameter and ΔE values determined? The manuscript lacks a detailed description of how it was determined.
Minor Comments:
1) Why is the delta-E notation used once and then ΔE?
2) The caption of Fig.2 lacks a description of the meaning of the arrow and the rectangle used.
3) Correct unnecessary repetitions (see, e.g., line 264: ‘synthesized as reported as reported”)
4) Provide direct specifications of used (version, etc.).
5) Quality Fig. 3 is too low.
Round 2
Reviewer 1 Report
I am happy that the authors have addressed all the comments raised. The manuscript now deserves to be published in Molecules. Best wishes for your publication in a reputed journal.
Reviewer 2 Report
I would like to thank you for the clarifications and changes made by the authors. The scientific quality of the manuscript has improved considerably. In my opinion, the current version of the manuscript is suitable for publication.